# Sprint Mechanical Properties of Female and Different Aged Male Top-Level German Soccer Players

**DOI:** 10.3390/sports6040161

**Published:** 2018-11-28

**Authors:** Christian Baumgart, Jürgen Freiwald, Matthias Wilhelm Hoppe

**Affiliations:** 1Department of Movement and Training Science, University of Wuppertal, Fuhlrottstraße 10, 42119 Wuppertal, Germany; freiwald@uni-wuppertal.de (J.F.), m.hoppe@uni-wuppertal.de (M.W.H.); 2Department of Orthopedic, Trauma and Hand Surgery, Klinikum Osnabrück GmbH, Am Finkenhügel 1, 49076 Osnabrück, Germany

**Keywords:** acceleration, elite players, football, running, speed

## Abstract

This study compared the sprint mechanical properties of female and different aged male top-level soccer players. A total of 14 adult females (FEM) and 115 different aged male field players, competing at German top levels, participated in this study. The males belonged to teams of under 12, 13, 14, 15, 17, 19, and 23 years (U 12–23) and professionals (PRO). All players were tested for a 30 m linear sprint. From timing gate derived sprint times, force-velocity and power-velocity relationships, as well as theoretical maximum running velocity, force, and power data were computed by an inverse dynamic approach applied to the center of mass. The approach was optimized for taking the starting time into account, which is a progress in the present research field, when aiming to compute sprint mechanical properties by different methodological approaches under field conditions. Sprint mechanical properties of FEM were lower than those of PRO. Compared to other age groups, sprint mechanical properties of FEM were similar to those of U 14 and U 15. An increase in sprint mechanical properties was found from U 12 to U 17. The study shows that sprint mechanical properties differ according to gender and age in top-level soccer players.

## 1. Introduction

In soccer, accelerated sprint running is a fundamental part of soccer performance and occurs frequently during matches [1]. Differences in sprint performances between female and male, as well as different aged soccer players, have been investigated using timing gate measurements in numerous previous studies [2,3,4,5]. Overall, the studies show that 1st division male players sprint faster than female players [4,6]. Additionally, during maturation, the sprint performance of young soccer players increases until the age of 17 [3,7], while a further increase beyond that age is controversially discussed [2,8,9]. Such knowledge can be used to profile the players concerning their sprint performance, which may helpful to design training drills and assist the talent selection processes.

To accelerate the body in a forward direction, high horizontal ground reaction forces are required [10]. Since it is not practical to measure such forces by force plates, a biomechanical model was recently developed [11]. The model is based on an inverse dynamic approach applied to the center of mass (COM) during maximal sprinting using timing gate, anthropometric, and environmental data as inputs. Thereby, the estimation of the theoretical horizontal force and power during accelerated sprint running under field conditions can be considered as valid and reliable when, at least in part, three split times are used [11,12]. The resulting data can be used to compute force-velocity and power-velocity relationships, reflecting the entire neuromuscular capacities to produce external force and power in the horizontal plane [13]. Compared to sprint times, knowledge of such mechanical profiles may allow more individualized training drills, because some underlying factors of the sprint performance are derived.

In soccer, sprint mechanical properties have been reported concerning the effects of training procedures [14], match play induced fatigue [15], and a hamstring injury [16]. Additionally, one study investigated the relationships between sprint mechanical properties and sprint times in young soccer players [17]. Overall, the findings indicate that sprint mechanical properties explain different components of the sprint performance in young soccer players. Therefore, and due to genetic, growth, development, and maturation effects [18], it can be expected that sprint mechanical properties differ between different aged male soccer players, which has not been investigated so far. Moreover, gender differences in sprint mechanical properties have only been reported in world-class 100 m sprinters [19], but not in soccer yet. Such knowledge may be useful to design more gender and age specific training drills to optimize the sprinting performance of soccer players.

This study aimed to compare the sprint mechanical properties of female and different aged male top-level soccer players. We hypothesized that the sprint mechanical properties increase with increasing age, and female players exhibit lower values than male adult players.

## 2. Materials and Methods

A total of 14 adult female (FEM) and 115 different aged male field soccer players participated. The males belonged to teams of under 12, 13, 14, 15, 17, 19 and 23 years (U 12–23), and professionals (PRO) of a 1st division German Bundesliga club. The FEM were also members of a 1st division German Bundesliga club. The training regime of the FEM consisted of four sessions per week, which is comparable to the U 12 to U 15. However, the U 17 to PRO trained on a daily basis, including up to two sessions per day. Over the entire season, all groups had at least one friendly or competitive match per week. At the time of testing, all players were free of acute injuries and illness. They were familiar with the testing procedures as a part of their regular performance assessment program. All players were informed about the study procedures and have given their written consent to participate. Parental consent was given for players under 18 years of age. All procedures were pre-approved by the Ethics Committee of the local university (MS/JE 29.11.11) and were conducted in accordance with the current laws and the Declaration of Helsinki.

### 2.1. Procedures

To compare the sprint mechanical properties of female and different aged male top-level soccer players, a descriptive cross-sectional study design was applied. First, all players were tested for body mass and height. Then, they performed a 15–20 min standardized warm-up, which was conducted by their coaches. After this, each player completed three 30 m linear sprints with maximal effort on artificial turf, as previously described [20]. Between each trial, a rest interval of three minutes was maintained. The sprinting track was equipped with double-light timing gates (Werthner Sport Consulting, TDS, Linz, Austria) positioned at a height of 85 and 105 cm. For the starting procedure, the rear foot was placed on a contact mat, whereas the front foot was placed behind the starting line. After authorization from an investigator, the players were allowed to independently start each trial. The sprint times were measured at 5, 10, 20, and 30 m.

### 2.2. Data Analysis

To increase the reliability, all data processing procedures were applied to the mean of the two fastest 30 m sprints [21]. All calculations were performed with custom-made spreadsheets using macro-based calculations (Microsoft, Excel 2016, Redmond, WA, USA). First, for each player, the distance-time function was fitted based on Equation (1), as described previously [11,22]:(1)$$x\left(t\right)=v_{max}\cdot \left(\left(t+t_{0}\right)-\tau +\tau \cdot e^{-\;\frac{t+t_{0}}{\tau }}\right),$$
where *v_max_* is the theoretical horizontal maximum running velocity reached at the end of the acceleration phase and *τ* is a time-constant. An individual starting-time (*t*_0_) was added to the measured sprint times to account for the starting procedure using a contact mat, on which the players produce a horizontal force impulse before the rear foot has left the ground. This represents a progress in the present research field, when aiming to compute sprint mechanical properties by different methodological approaches under field conditions. Accordingly, the *v_max_*, *τ*, and *t*_0_ were determined by the least-square regression method using the known distances and corresponding sprint times as inputs. The mean *v_max_*, *τ*, and *t*_0_ for all players were 7.96 ± 0.84 m/s, 1.10 ± 0.11 s, and 0.27 ± 0.05 s, respectively.

Then, the theoretical acceleration- and velocity-time data were calculated from 0 s to 5 s with the following functions [11]:(2)$$a\left(t\right)=\left(\frac{v_{max}}{\tau }\right)\cdot e^{-\;\frac{t}{\tau }},$$

(3)$$v\left(t\right)=v_{max}\cdot \left(1-e^{-\;\frac{t}{\tau }}\right).$$

Thereon, the theoretical horizontal force-time data were determined accordingly:(4)$$F\left(t\right)=m\cdot a\left(t\right)+F_{aero}\left(t\right),$$
where *m* is the measured body mass, *a*(*t*) the acceleration from Equation (2), and *F_aero_* is the required aerodynamic drag force to overcome during sprinting, which was calculated based on body height and environmental conditions as described previously [11]. Lastly, the theoretical horizontal power was computed as [11]:(5)P(t)=F(t)·v(t).

Based on the computed data, force-velocity and power-velocity relationships were plotted. The theoretical horizontal maximum running velocity (*v_max_*), force (*F_max_*), power (*P_max_*), and velocity at *P_max_* (*v_Pmax_*) were used for statistical analyses. To detect potential effects of body mass, all force and power data were analyzed in absolute and relative units.

### 2.3. Statistical Analysis

Descriptive statistics were reported as mean values and standard deviations (sd). Since the normal distribution within some groups was not given, non-parametric tests were consistently applied. The Kruskal–Wallis test was used to evaluate group differences. A level of *p* < 0.05 was set for global statistical significance. Wilcoxon signed-rank tests were applied to determine post-hoc differences with an adjusted significance level set at *p* ≤ 0.002. The SPSS software (IBM, Version 24, Armonk, NY, USA) was used for all statistical calculations.

## 3. Results

For all investigated groups, the calculated sprint mechanical properties are summarized in Table 1. The anthropometric characteristics and sprint times are also shown. Age significantly increases from U 12 to the adult groups. No significant differences were found between FEM, PRO, and U 23. For all groups, the mean velocity-time, force-velocity, and power-velocity relationships are presented in Figure 1.

### 3.1. Gender Differences

With exception of the relative F_max_, sprint mechanical properties of FEM were lower than those of U 17 to PRO, and higher than those of U 12. Moreover, the F_max_ and P_max_ of FEM were higher than those of U 13 and U 14. Additionally, the v_max_ of FEM was lower than that of U 15 (Table 1 and Figure 2).

### 3.2. Age Related Differences within Males

The sprint mechanical properties of U 17 were higher than those of U 12 to U 14 (v_max_, v_Pmax_, relative F_max_, and relative P_max_) and U 12 to U 15 (F_max_ and P_max_). The sprint mechanical properties of the PRO were higher than those of U 12 to U 15 (v_max_, v_Pmax_, F_max_, and P_max_) and U 12 to U 14 (relative F_max_ and relative P_max_) (Table 1). No significant group differences in the sprint mechanical properties were present from U 17 to PRO. However, the mean absolute values of F_max_ and P_max_ increased, while relative to body mass a stagnation occurred (Figure 2).

## 4. Discussion

This study was the first to compare sprint mechanical properties of female and different aged male top-level German soccer players. Our main findings were: (1) sprint mechanical properties of FEM were lower than those of PRO and comparable to those of male U 14 and U 15 players; and (2) sprint mechanical properties of male players increased from U 12 to U 17.

### 4.1. Gender Differences

Our first major finding was that sprint mechanical properties of FEM were lower than those of PRO (Table 1 and Figure 2). Since our study was the first to investigate such gender differences in soccer players, no comparisons with previous studies are possible. However, our gender differences in soccer players (v_max_: −17%; F_max_: −31%; P_max_: −43%; relative P_max_: −21%) were comparable to those reported by a previous study that has applied the same inverse dynamic approach for world-class 100 m sprinters (v_max_: −10%; F_max_: −32%; P_max_: −38%; relative P_max_: −19%) [19], which supports our computations. Noteworthy, our results shows that gender differences in soccer players increase more at higher sprinting velocities (Figure 1) than those of the 100 m sprinters [19]. Thus, the capacity to produce high horizontal impulses (forces) at higher velocities could be a promising training goal in female soccer players. In fact, in professional female players, assisted and resisted sprint training enhanced acceleration and maximum sprint velocity, respectively [23]. In general, gender differences in sport are related to genetic (e.g., anthropometric and biomechanical aspects, and trainability) and environmental factors (e.g., training content and conditions) [18,24,25]. Since environmental factors in top-level soccer and world-class sprinters are comparable within and between genders, and our differences in soccer players were only slightly higher than those in sprinters, our gender differences may be mainly caused by genetic factors [25].

Concerning gender differences, our study shows that sprint mechanical properties of FEM players were comparable to those of male U 14 and U 15 players (Table 1 and Figure 2). Cautiously, this finding is supported by the results of friendly matches played between female and male junior soccer players. Indeed, female national teams lose against male U 15 teams (2016 AUS vs. Newcastle 0:7; 2017 USA vs. Dallas 2:5). Overall, the findings may suggest that there is a relationship between sprint mechanical properties and results of friendly matches played between adult female and male junior soccer players.

### 4.2. Age Related Differences within Males

The second main finding was that group differences in the mechanical properties were found from U 12 to U 17, whereby the younger players showed lower values than the older players. No significant differences were present between players from U 17 to PRO. Compared to our PRO, only one previous study has also investigated sprint mechanical properties in soccer players and reported similar v_max_ (8.86 m/s), but lower relative F_max_ (6.8 N/kg) and P_max_ (15.0 W/kg) data for semi-professional players [16]. These data again support our computations. No further previous studies in soccer players are available. However, for international rugby players also similar v_max_ (9.03 m/s), but slightly higher relative F_max_ (8.84 N/kg) and P_max_ (19.9 W/kg) compared to our PRO players have been reported [26]. Moreover, the calculated theoretical maximum sprinting velocity of PRO (8.9 m/s) was in line with that reported for elite soccer players before (8.8-9.0 m/s) [8].

It is well known that naturally occurring growth, development, and maturation processes (e.g., induced by increased testosterone levels) as well as training adaptations according to sprint, resistance, or plyometric drills (e.g., increased muscle mass and neuronal control) can enhance relative force and power generating capacities [18]. These factors may explain our increased sprint mechanical properties from U 12 to U 17. However, Figure 2 shows that the mean absolute P_max_ values increased from U 17 to PRO, but stagnated relative to body mass. This outcome is supported by our data (Table 1) and numerous previous studies [2,3,7,8]. Overall, sprinting performance increased until U 17 and stagnated thereafter. Furthermore, while body mass increases from adolescents to adults [3,8,9], relative to body mass, the muscle mass and maximum strength capacities did not differ between adolescent and adult players [3,9]. In our study, the relative force- and power-values (Figure 1) of the PRO players were the highest at higher velocities (>5 m/s) compared to the other group. Beside the aforementioned aspects, further factors, particularly changes in connective tissue properties, may also play a role for differences in sprint mechanical properties. For example, during late adolescence, a pronounced tendon hypertrophy in relation to the functional and morphological development of the muscle occurs [27].

From a practical point of view, not only sprint times, as shown before [2], but also relative sprint mechanical properties (Table 1 and Figure 2) have limited power to distinguish adolescents from PRO soccer players. This should be considered by coaches, when aiming to test sprint performances.

### 4.3. Methodological Aspects

Finally, due to their increasing scientific and practical attention [10], some methodological aspects of sprint mechanical properties are worth discussing. Several authors proposed that sprint mechanical properties are useful for an explanation of different components of sprint performance [17,28]. However, it is important to understand that the computation of sprint mechanical properties is based on the assessment and modeling of a velocity-time curve [11]. This velocity-time origin and our findings collectively show that differences in sprint mechanical properties and sprint times of soccer players go ‘hand in hand’ (Table 1 and Figure 1). Thus, it is presently unknown if sprint mechanical properties are more useful for practical applications (e.g., profiling, training, or talent selection) than traditional sprint time analyses. 

Besides these concerns, two major advantages of sprint mechanical properties should be highlighted. First, exercises performed in horizontal (i.e., accelerated sprint running) and vertical planes (e.g., jumps, bench press, and squat exercises) can be compared due to the same measurement unit (i.e., force and power) [29]. This may allow better profiling and more individualized training of soccer players. A second advantage of sprint mechanical properties may be related to the modeling procedures of the velocity-time curve. While traditional sprint time analyses are affected by several methodological aspects (e.g., sprinting distances, starting positions, and types of timing gates), leading to outcomes that are difficult to compare [30,31], the impact of such issues may be reduced by the modeling of a velocity-time curve [12]. If so, then sprint mechanical properties may allow a better comparison between studies, being especially important for meta-analyses. Lastly, with the modelling in mind, a methodological strength of our study was that we have optimized the inverse dynamic approach for computing sprint mechanical properties by taking the exact starting time into account. This progress reduces the impact of different starting procedures (e.g., flying start or contact mat) and measurement devices (e.g., timing gates, laser, radar, or global and local positioning systems) on sprint mechanical property outcomes [22]. Generally, the use of mechanical properties may become more important in the future and were not limited to standardized sprint tests, as investigated here, but may also be useful for estimating external loads of intermittent running activities in team and racquet sports during match play or training drills [32].

### 4.4. Limitations

While our study increased knowledge regarding sprint mechanical properties in top-level German soccer players, our results were limited by the circumstance that we applied a descriptive cross-sectional study design. Consequently, whether our differences between different aged players were based on growth, development, and maturation or training and selection remain unknown [33]. Moreover, the applied approach of sprint mechanical properties only accounts for the power of the horizontal acceleration of the COM. Since the COM is also accelerated in a vertical direction during each step while sprinting, potential gender and age-related effects concerning this aspect were not considered. Lastly, we included only one adult team of FEM, and therefore, age-related aspects in female soccer players remain unknown.

## 5. Conclusions

This study shows that sprint mechanical properties differ according to gender and age in top-level German soccer players. An increase in sprint mechanical properties was found from U 12 to U 17. Sprint mechanical properties of FEM were lower than those of PRO, and similar to those of U 14 and U 15. More research is required to determine, if sprint mechanical properties allow a better profiling of top-level soccer players concerning their sprint performance, potentially permitting more individualized training drills.

## Figures and Tables

**Figure 1 sports-06-00161-f001:**
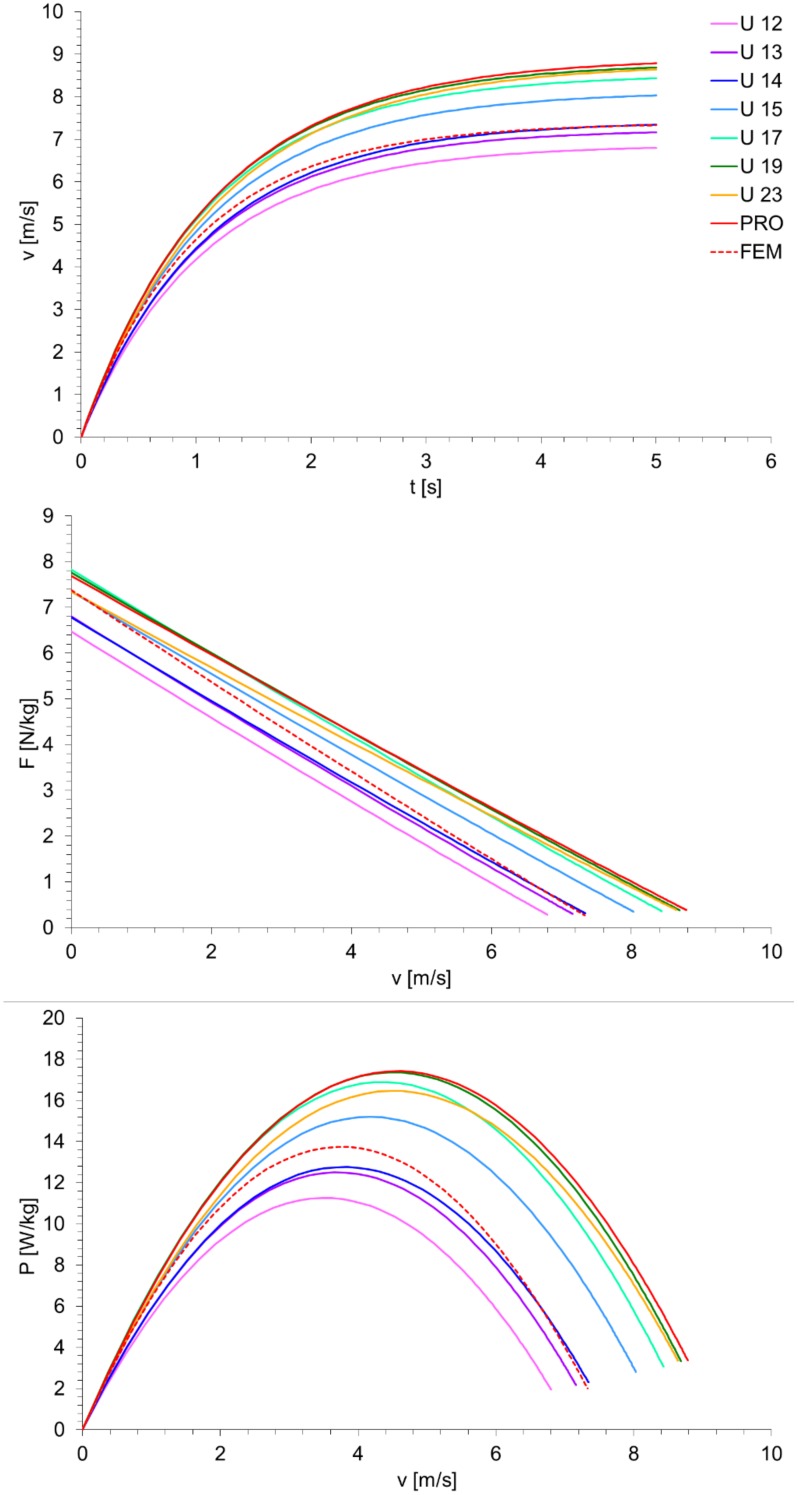
Mean velocity-time, force-velocity, and power-velocity relationships of female and different aged male top-level German soccer players.

**Figure 2 sports-06-00161-f002:**
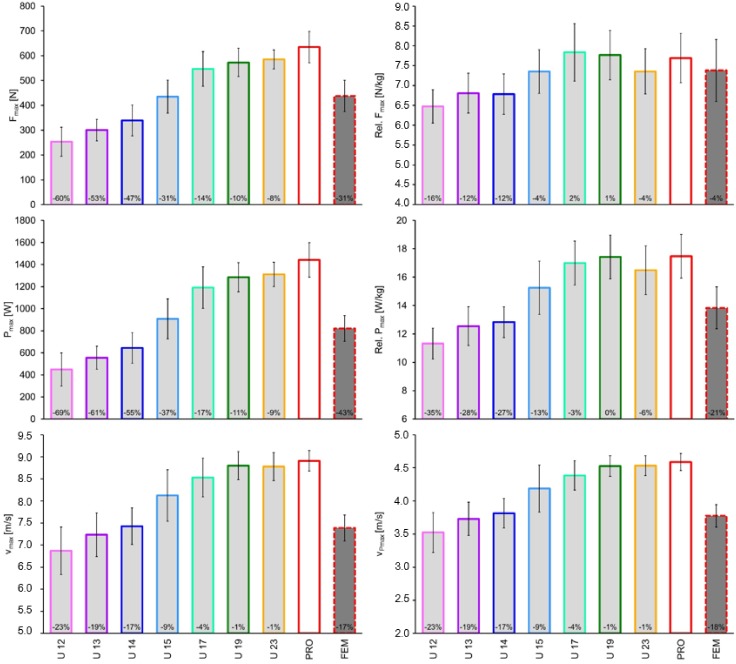
Maximum absolute and relative force (F_max_) and power (P_max_) data of female and different aged male top-level German soccer players. The theoretical maximum running velocity (v_max_) and velocity at maximum power (v_Pmax_) are also shown. The percentage differences were computed for each group in relation to the male professional players (PRO).

**Table 1 sports-06-00161-t001:** Anthropometric characteristic, sprint times, and sprint mechanical properties of female and different aged male top-level German soccer player (mean ± sd).

Parameter	U 12 (n = 15)	U 13 (n = 17)	U 14 (n = 16)	U 15 (n = 13)	U 17 (n = 14)	U 19 (n = 13)	U 23 (n = 14)	PRO (n = 13)	FEM (n = 14)	Kruskal-Wallis Test
Age [year]	11.6 ± 0.3 ^all^	12.6 ± 0.3 ^all^	13.6 ± 0.3 ^all^	14.6 ± 0.2 ^all^	16.4 ± 0.6 ^all^	18.1 ± 0.7 ^all^	22.1 ± 5.2 ^12;13;14;15;17;19^	25.7 ± 4.7 ^12;13;14;15;17;19^	23.2 ± 4.2 ^12;13;14;15;17;19^	*χ*^2^ = 122.2 *p* < 0.001
Mass [kg]	39.2 ± 9.1 ^all^	44.1 ± 5.1 ^12;15;17;19;23;P;F^	49.9 ± 7.6 ^12;15;17;19;23;P;F^	59.0 ± 6.8 ^12;13;14;19;23;P^	70.0 ± 8.7 ^12;13;14;P;F^	73.7 ± 4.0 ^12;13;14;15;P;F^	79.8 ± 5.8 ^12;13;14;15;F^	82.6 ± 7.3 ^12;13;14;15;17;19;F^	59.3 ± 4.8 ^12;13;14;17;19;23;P^	*χ*^2^ = 107.7 *p* < 0.001
Height [m]	1.50 ± 0.09 ^14;15;17;19;23;P;F^	1.55 ± 0.06 ^15;17;19;23;P;F^	1.62 ± 0.08 ^12;17;19;23;P^	1.69 ± 0.07 ^12;13;17;19;23;P^	1.79 ± 0.07 ^12;13;14;15;F^	1.79 ± 0.05 ^12;13;14;15;F^	1.83 ± 0.06 ^12;13;14;15;F^	1.85 ± 0.06 ^12;13;14;15;F^	1.66 ± 0.06 ^12;13;17;19;23;P^	*χ*^2^ = 98.8 *p* < 0.001
t–5 m [s]	1.26 ± 0.04 ^14;15;17;19;23;P^	1.25 ± 0.06 ^15;17;19;23;P^	1.19 ± 0.05 ^12;17;19;23;P^	1.16 ± 0.03 ^12;13;17;19;23;P^	1.10 ± 0.03 ^12;13;14;15;F^	1.12 ± 0.03 ^12;13;14;15;F^	1.10 ± 0.03 ^12;13;14;15;F^	1.11 ± 0.03 ^12;13;14;15;F^	1.21 ± 0.09 ^17;19;23;P^	*χ*^2^ = 96.0 *p* < 0.001
t–10 m [s]	2.14 ± 0.07 ^14;15;17;19;23;P;F^	2.09 ± 0.10 ^15;17;19;23;P^	2.02 ± 0.07 ^12;17;19;23;P^	1.94 ± 0.07 ^12;13;17;19;23;P^	1.84 ± 0.05 ^12;13;14;15;F^	1.85 ± 0.04 ^12;13;14;15;F^	1.84 ± 0.04 ^12;13;14;15;F^	1.84 ± 0.05 ^12;13;14;15;F^	2.03 ± 0.10 ^12;17;19;23;P^	*χ*^2^ = 100.0 *p* < 0.001
t–20 m [s]	3.68 ± 0.14 ^14;15;17;19;23;P;F^	3.56 ± 0.17 ^15;17;19;23;P^	3.47 ± 0.12 ^12;15;17;19;23;P^	3.28 ± 0.14 ^12;13;14;17;19;23;P^	3.13 ± 0.09 ^12;13;14;15;F^	3.11 ± 0.07 ^12;13;14;15;F^	3.11 ± 0.08 ^12;13;14;15;F^	3.09 ± 0.08 ^12;13;14;15;F^	3.46 ± 0.14 ^12;17;19;23;P^	*χ*^2^ = 100.9 *p* < 0.001
t–30 m [s]	5.17 ± 0.23 ^14;15;17;19;23;P;F^	4.98 ± 0.26 ^15;17;19;23;P^	4.84 ± 0.18 ^12;15;17;19;23;P^	4.55 ± 0.22 ^12;13;14;19;23;P^	4.33 ± 0.14 ^12;13;14;F^	4.28 ± 0.11 ^12;13;14;15;F^	4.30 ± 0.12 ^12;13;14;15;F^	4.25 ± 0.10 ^12;13;14;15;F^	4.84 ± 0.19 ^12;17;19;23;P^	*χ*^2^ = 99.8 *p* < 0.001
v_max_ [m/s]	6.87 ± 0.54 ^14;15;17;19;23;P;F^	7.23 ± 0.49 ^15;17;19;23;P^	7.43 ± 0.41 ^12;17;19;23;P^	8.12 ± 0.58 ^12;13;P;F^	8.53 ± 0.44 ^12;13;14;F^	8.80 ± 0.32 ^12;13;14;F^	8.78 ± 0.32 ^12;13;14;F^	8.91 ± 0.23 ^12;13;14;15;F^	7.39 ± 0.30 ^12;15;17;19;23;P^	*χ*^2^ = 96.1 *p* < 0.001
F_max_ [N]	253 ± 58 ^all^	300 ± 43 ^12;15;17;19;23;P;F^	339 ± 62 ^12;17;19;23;P;F^	435 ± 67 ^12;13;17;19;23;P^	547 ± 70 ^12;13;14;15;F^	573 ± 57 ^12;13;14;15;F^	585 ± 38 ^12;13;14;15;F^	635 ± 63 ^12;13;14;15;F^	438 ± 63 ^12;13;14;17;19;23;P^	*χ*^2^ = 107.5 *p* < 0.001
F_max_ [N/kg]	6.5 ± 0.4 ^15;17;19;23;P;F^	6.8 ± 0.5 ^17;19;P^	6.8 ± 0.5 ^17;19;P^	7.4 ± 0.6	7.8 ± 0.7 ^12;13;14^	7.8 ± 0.6 ^12;13;14^	7.4 ± 0.6	7.7 ± 0.6 ^12;13;14^	7.4 ± 0.8	*χ*^2^ = 53.3 *p* < 0.001
P_max_ [W]	449 ± 150 ^all^	555 ± 105 ^12;15;17;19;23;P;F^	644 ± 138 ^12;15;17;19;23;P;F^	907 ± 181 ^12;13;14;17;19;23;P^	1191 ± 188 ^12;13;14;15;F^	1284 ± 132 ^12;13;14;15;F^	1311 ± 109 ^12;13;14;15;F^	1442 ± 156 ^12;13;14;15;F^	821 ± 116 ^12;13;14;17;19;23;P^	*χ*^2^ = 109.5 *p* < 0.001
P_max_ [W/kg]	11.3 ± 1.1 ^14;15;17;19;23;P;F^	12.5 ± 1.4 ^15;17;19;23;P^	12.8 ± 1.1 ^12;15;17;19;23;P^	15.2 ± 1.9 ^12;13;14^	17.0 ± 1.5 ^12;13;14;F^	17.4 ± 1.5 ^12;13;14;F^	16.5 ± 1.7 ^12;13;14;F^	17.5 ± 1.5 ^12;13;14;F^	13.8 ± 1.5 ^12;17;19;23;P^	*χ*^2^ = 92.5 *p* < 0.001
v_Pmax_ [m/s]	3.52 ± 0.30 ^14;15;17;19;23;P;F^	3.73 ± 0.25 ^15;17;19;23;P^	3.82 ± 0.22 ^12;17;19;23;P^	4.19 ± 0.35 ^12;13;P^	4.39 ± 0.22 ^12;13;14;F^	4.53 ± 0.16 ^12;13;14;F^	4.53 ± 0.15 ^12;13;14;F^	4.59 ± 0.13 ^12;13;14;15;F^	3.77 ± 0.17 ^12;17;19;23;P^	*χ*^2^ = 93.4 *p* < 0.001

Note. v_max_—theoretical maximum running velocity; F_max_—theoretical maximum horizontal force; P_max_—theoretical maximum mechanical power; v_Pmax_—theoretical velocity at P_max_. Significant (*p* ≤ 0.002) post-hoc Wilcoxon signed-rank tests were marked as 12–23 (U 12–U 23), P—professionals (PRO), and F—females (FEM).

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
