# Peer review of "Sprint Mechanical Properties of Female and Different Aged Male Top-Level German Soccer Players"

_sports, 2018, doi:10.3390/sports6040161_

Reviewer 1 Report

GENERAL COMMENTS

This study reports data for maximum velocity, maximum force, and maximum power in sprinting by German football players. First devision male players, first division female players, and several sets of age group male players were tested.

There are two parts to the study reported here.

1. Comparing female first divison players to male first division players.

2. Comparing male age group players to male first division male players.

Need to more clearly separate these in the Introduction, Methods, Results, and Discussion.

Writing

The writing could be substantially improved. Every section need work in improving the clarity of expression.

There are many instances of cumbersome groupings of words. For example:

‘timing-gate derived sprint analyses’, ‘timing-gate derived knowledge’, ‘mechanical properties based profiles’, ‘sprint performance underlying factors’, ‘to detect body mass driven effects’.

Please do not use manufactured or unfamiliar acronyms (e.g SMP, FEM, PRO); they make the manuscript difficult to follow.

You should find the following books very useful:

Cutts, M. (2013). Oxford guide to plain English (4th ed.). Oxford: Oxford University Press.

De A’Morelli, R. (2016). Elements of style 2017. Vancouver: Spectrum Ink.

Kirkman, J. (2006). Good style: Writing for science and technology (2nd ed.). London: Routledge.

The following article is a good guide to preparing a manuscript:

Brand, R. A., & Huiskes, R. (2001). Structural outline of an archival paper for Journal of Biomechanics, Journal of Biomechanics, 34, 1371-1374.

You may find the following book useful when preparing a manuscript:

Day, R., & Gastel, B. (2012). How to write and publish a scientific paper (7th Ed.), Cambridge: Cambridge University Press.

You may find the following book useful when producing graphs:

Cleveland, W. S. (1994). The elements of graphing data. Summit, NJ: Hobart Press.

SPECIFIC COMMENTS

Introduction

The reader is unlikely to know what you mean by ‘sprint mechanical properties’.

Line 27 ‘…a fundamental part of the multifactorial performance.’ This is vague. What do you mean?

Line 42: The power-velocity curve is not formed from the equation of a parabola. Better to describe it as an inverted u-shape.

Methods

Equation 1 is not correct. There is a missing negative sign.

Line 88-92: Suggested change: ‘We added 0.27 s to the start time to account for the time delay between when the participant commenced the sprint and the time when the rear foot broke contact with the contact mat.’ Have I got this right? Did you apply a different time correction to each player? If so, how was this decided? (Explain in one sentence.)

Line 94: Is this the mean of all participants in the study. If so, what is it telling me?

Line 102: The power-velocity curve is not formed from the equation of a parabola.

Results

Line 123-126: I cannot make sense of this. The message is not clear to me.

Line 128-133: I cannot make sense of this. The message is not clear to me.

Table 1: It looks like the ages were recorded to the nearest year, and so the standard deviation is zero for the age group players. The resolution of the age measurements (to the nearest year) should be mentioned in the methods.

Do we really need this table of data? The data is also in Figure 2. Table 1 could be an appendix or Supplementary Material. Do we really need to know if there are significant differences? Many statisticians recommend reporting differences and effect sizes, rather than concentrating on statistical significance.

Figure 1: The labels should be ‘Men 1st division’, ‘Women 1st division’, ‘Boys Under 12’, etc

Discussion

This section is too long. There is not much to say beyond the good summary in the first paragraph.

Line 151-168: This is not clear. The message does not appear to be important and should be deleted.

Line 173-197. The message does not appear to be important and should be deleted.

Author Response

We have added our reply in a pdf file.

Reviewer 2 Report

Overall, the manuscript is well written and data that was collected is clearly presented. However, there are a few minor questions/suggestions:

It is mentioned that there was a single female group of 14 adult females. However, compared to the male groups, which have very specified age ranges, there is no description regarding the age criteria within this female group. In the table, I noticed that an average age with standard deviation is listed, but no criteria for this specific population was defined. Since this is the only female group, greater description of this demographic is needed. 

Is there a reason why only a single female group was used for data collection vs many different groups for the male populations? Is there a shortage of female soccer teams in the authors region to draw a comparison from? A gender discrepancy would hold much greater validity if additional female groups were used for proper comparison as opposed to the 1 female group vs 8 male groups utilized in this current study. This does not necessarily need to be expanded for publication purposes, but this needs to be addressed in the methodologies and the limitations of the study. 

Although the graphs in Figure 1 are very helpful for demonstrating overall performance metrics, it is very difficult to distinguish which group is which when looking at these graphs in black and white. If possible, assigning colors to these individual line segments vs gray scale would be very helpful to the reader.  

Author Response

We have added our reply in a pdf file.
